# Perceived barriers to cervical cancer screening and motivators for at-home human papillomavirus self-sampling during the COVID-19 pandemic: Results from a telephone survey

Susan Parker[1]*, Ashish A Deshmukh[2], Baojiang Chen[3], David R Lairson[3], Maria Daheri[4], Sally W Vernon[3], Jane R Montealegre[1]

[1]Baylor College of Medicine, Houston, United States; [2]Medical University of South Carolina, Charleston, United States; [3]UTHealth School of Public Health, Houston, United States; [4]Harris Health System, Houston, United States

*For correspondence: susan.parker2@bcm.edu

Competing interest: The authors declare that no competing interests exist.

## Abstract

**Background:** Home-based self-sampling for human papillomavirus (HPV) testing may be an alternative for women not attending clinic-based cervical cancer screening.

**Methods:** We assessed barriers to care and motivators to use at-home HPV self-sampling kits during the COVID-19 pandemic as part of a randomized controlled trial evaluating kit effectiveness. Participants were women aged 30–65 and under-screened for cervical cancer in a safety-net healthcare system. We conducted telephone surveys in English/Spanish among a subgroup of trial participants, assessed differences between groups, and determined statistical significance at p<0.05.

**Results:** Over half of 233 survey participants reported that clinic-based screening (Pap) is uncomfortable (67.8%), embarrassing (52.4%), and discomfort seeing male providers (63.1%). The last two factors were significantly more prevalent among Spanish vs English speakers (66.4% vs 30% (p=0.000) and 69.9 vs 52.2% (p=0.006), respectively). Most women who completed the kit found Pap more embarrassing (69.3%), stressful (55.6%), and less convenient (55.6%) than the kit. The first factor was more prevalent among Spanish vs English speakers (79.6% vs 53.38%, p=0.001) and among patients with elementary education or below.

**Conclusions:** The COVID-19 pandemic influenced most (59.5%) to participate in the trial due to fear of COVID, difficulty making appointments, and ease of using kits. HPV self-sampling kits may reduce barriers among under-screened women in a safety-net system.

**Funding:** This study is supported by a grant from the National Institute for Minority Health and Health Disparitie s (NIMHD, R01MD013715, PI: JR Montealegre).

**Clinical trial number:** NCT03898167.

## Editor's evaluation

The evidence presented in the manuscript is solid, and the study is a valuable contribution to research on at-home sampling for cervical cancer screening in underserved populations. The fact that the study was conducted during the COVID-19 pandemic makes it particularly informative for policymaking in circumstances of restricted access to care.

## Introduction

The disruptions in the US healthcare system due to the COVID-19 pandemic have resulted in a sharp decline in routine primary care, including cervical cancer screening (*Czeisler et al., 2020*). This is expected to lead to gaps in preventive care and increased risk of preventable chronic diseases (*Wright et al., 2020*; *CDC, 2020*), especially among medically underserved populations. Cervical cancer screening declined by 84% in April 2020 (*DeGroff et al., 2021*), a month after the declaration of the global COVID-19 pandemic, and the rates had not yet fully recovered by June 2021 (*Mast et al., 2021*). Before the COVID-19 pandemic, racial minorities and those with limited English proficiency were less likely to be screened for cervical cancer than their non-Hispanic white and English-proficient counterparts (*Fuzzell et al., 2021*), leading to disparities in cervical cancer incidence and mortality (*National center for health statistics and National Health Interview Survey, 2019*). These populations experiencing higher rates of cervical cancer and other chronic illnesses before the pandemic are now faced with widening health disparities due to COVID (*Fisher-Borne et al., 2021*).

Safety net health systems, which provide care regardless of the patient's ability to pay, provide care for a large proportion of the medically underserved population in the US and have become increasingly important during the COVID-19 pandemic (*Knudsen and Chokshi, 2021*). The population served by safety net systems predominantly comprises low-income individuals, immigrants, and racial/ethnic minorities (*America's Health Care Safety Net, 2000*). These populations are also disproportionately affected by COVID (*Mullangi et al., 2020*).

Barriers to cervical cancer screening among safety net system patients, both pre- and post–pandemic, have not been fully described, and thus research to inform targeted approaches to increase screening participation is needed. A previous study found that under-screened women within a safety net system were more likely to have limited knowledge of HPV and report cost, time, and lack of childcare as barriers to Pap screening compared to screened women (*Ogunwale et al., 2016*). In this context, alternative screening strategies such as home-based self-sampling for HPV testing may help circumvent many of these barriers. Additionally, other barriers introduced by the COVID-19 pandemic, such as such as limited availability of clinic appointments and fear of illness, are also addressed by home-based self-sampling, which may provide opportunities to continue to deliver preventive care during disruptions.

Self-sample HPV testing is effective at detecting high-risk HPV (*Herrington, 2022*) and has been used in multiple settings for cervical cancer screening (*Nishimura et al., 2023*; *Gupta et al., 2018*; *Lim et al., 2017*). Increased participation in screening varies across settings and by implementation strategies used, but a recent meta-analysis was associated with a nearly doubling of cervical cancer screening rates (*Musa et al., 2017*). Furthermore, self-sampling is highly acceptable by patients in multiple healthcare settings (*Nelson et al., 2017*). In the United States, the National Cancer Institute is currently conducting the 'Last Mile' initiative to provide data to support FDA approval of self-sampling (*National Cancer Institute. Division of Cancer Prevention, 2022*). If approved, self-sampling could be used in healthcare settings to address barriers to screening, particularly in safety net systems where screening coverage is generally low (*Bauer et al., 2022*). Thus it is imperative to understand current barriers to screening in safety net health systems, as well as motivators to use self-sample HPV testing.

Here, we describe perceived barriers to cervical cancer screening and motivators to use an at-home self-sampling kit for HPV testing among women in an urban safety net health system. The survey was conducted among a subset of participants from the PRESTIS trial, a pragmatic trial assessing the effectiveness of mailed self-sample HPV kits to improve cervical cancer screening among women in a safety net healthcare system (*Montealegre et al., 2020*). The trial was predominantly conducted during the period of COVID-19-related measures, thus providing unique data on barriers and motivators to self-sampling during the COVID-19 pandemic. Here, we describe how safety net patients were affected by COVID, their perceptions of how the COVID-19 pandemic affected their participation in the trial, as well as barriers to clinic-based screening and motivators to use the self-sampling kits.

## Materials and methods

### Participants

Study participants were part of a larger HPV self-sampling randomized clinical trial, the Prospective Evaluation of Self-Testing to Increase Screening (PRESTIS) study (*Montealegre et al., 2020*). The trial is being conducted in a large, urban safety net health system, Harris Health System, which is 54.1% Hispanic/Latino, 25.9% Black/African American, 11.3% non-Hispanic White, and 8.7% Asian or other (*Harris Health System, 2021*). The trial began in Febraury 2020, paused in March due to COVID-19-related closures, and resumed in August 2020 when COVID-19-related research restrictions were lifted. The trial's protocol has been described in detail elsewhere (*Montealegre et al., 2020*). Briefly, patients are eligible for PRESTIS if they meet the following inclusion criteria: (1) 30–65 years of age; (2) no history of hysterectomy or cervical cancer; (3) under-screened for cervical cancer (no Pap test in the past 3.5 years or Pap/HPV co-test in the past 5.5 years); (4) at least two visits within the safety net healthcare system in the past 3.5 years; and (5) currently enrolled in a healthcare coverage or financial assistance plan accepted by the system (including Medicaid/Medicare, private insurance, and county-sponsored coverage). The latter two criteria were used to ensure that participants are current users of the healthcare system. Eligible patients were randomized to one of three study arms: Arm (1) Telephone recall (control) with a reminder to schedule a Pap test; Arm (2) Telephone recall with mailed self-sampling kit for HPV testing (intervention); and Arm (3) Telephone recall with mailed HPV self-sampling kit and an additional reminder/educational call from a health system employee (intervention plus). The self-sampling kits included an Aptima Multitest Swab collection kit to be returned to the health system for HPV testing.

As part of the trial, we conducted a nested survey to assess acceptability and experiences among a subset of randomly selected trial participants randomized to home-based self-sampling for HPV testing. This study includes telephone survey participants who responded between August 2020 and September 2022. Telephone survey participants were a random sample of women selected from each of four categories based on two factors: receipt of patient navigation (yes [Arm 3] or no [Arm 2]) and kit completed and returned within 6 months of randomization (yes or no). Women who require clinical follow-up were not eligible for this survey.

### Data collection

The survey was administered by trained, bilingual researcher coordinators in the patient's preferred language (English or Spanish). Participants were asked to provide verbal consent before commencing the survey and were sent a $20 gift card upon completion. This research was reviewed and approved by Baylor College of Medicine and Harris Health System's Institutional Review Boards (H-44944).

### Measures

The telephone survey was based on a questionnaire used in a previous study (*Montealegre et al., 2015*). Questions assess healthcare access and utilization (including specific questions about experiences during COVID-19-related closures and restrictions), barriers to cervical cancer screening, demographics, and telehealth access. Barriers to clinic-based screening were adapted from existing validated instruments (*Nelson et al., 2017*; *Byrd et al., 2007*; *Byrd et al., 2004*) and assessed using an 18-item scale, with items such as 'I don't have time to get a Pap test' and 'It's difficult to get an appointment for a Pap test.' Responses were on a three-point Likert scale (not at all, a little, very much) with an 'unsure/cannot say' option. Motivators were assessed by asking participants who reported using the kit to compare the convenience, stress/anxiety, and embarrassment of a Pap and the at-home self-sample kit by selecting whether the Pap at a clinic is more convenient/stressful/embarrassing, the self-sampling kit is more convenient/stressful/embarrassing, or the two screening methods are about the same. The motivators (convenience, stress, and embarrassment) of using the at-home kit vs clinic-based sampling were assessed with individual questions.

We assessed COVID-related experiences among all survey participants by asking whether the pandemic affected their economic situation, mental health, and physical well-being. Responses were on a 3-point Likert scale (large effect, small effect, no effect). To assess the influence of the COVID-19 pandemic, participants who reported using the kit were asked whether the COVID pandemic influenced their decision to participate in the trial. Those who indicated that the pandemic affected their decision were asked, 'In what way did the COVID-19 pandemic affect your participation in this trial?'

After thoroughly reading the recorded responses, the responses were coded using a grounded theory approach (*Glaser and Strauss, 1999*). Codes were then categorized into emerging themes.

## Analysis

Descriptive statistics were used to summarize the data. Chi-square or Fisher's exact tests for independence were conducted to assess the relationship between survey question responses and demographics. Fisher's exact test was used when more than 20% of cells had less than five participants, and chi-square was used for all other comparisons All statistical analyses were conducted using Stata IC 15.

## Results

A total of 233 telephone surveys were completed by patients enrolled in the PRESTIS study between August 2020 and September 2022. Most surveys (61.4%) were conducted in Spanish, and most participants (69.5%) were Hispanic/Latino, with the largest proportion (39.5%) born in Mexico (*Table 1*). Over 95% of participants who responded to the income and education questions had a total household income of less than $50,000, and 45.6% had less than a high school education, respectively. Spanish-speaking participants had significantly lower education completion levels than English-speaking participants (p=0.000).

### Self-reported barriers

The most commonly reported barriers to cervical cancer screening were a Pap being uncomfortable (67.8%) and the patient being uncomfortable with a male provider (63.1%). More Spanish-speaking participants reported being uncomfortable with a male provider as a barrier (69.9%) (*Table 2*) compared to English-speaking participants (52.2%, p=0.006) and Hispanic women were also significantly more likely to report this barrier than Black and White women (67.3% vs 51.0% and 42.9%, respectively, p=0.034). A similar pattern was seen among women who reported that getting a Pap is embarrassing (52.4% overall). Significantly more Spanish-speaking and Hispanic participants said that getting a Pap test is embarrassing compared to English-speaking and non-Hispanic participants (66.4% of Spanish speakers vs 30% of English speakers, p=0.000; 61.7% of Hispanic women vs 25.5% of Black, and 42.9% of White women, p=0.000). Participants with lower education were more likely to report embarrassment as a barrier (56.3% of elementary or less, 59.1% of high school-, and 36.1% of college-educated participants, p=0.021). Most women reported that getting a Pap test was not expensive (68.5%), with significantly more Spanish- vsEnglish-speaking women saying that getting a Pap is expensive (25.4% vs 12.2% for English-speaking participants, p=0.024). Most women reported that getting a Pap is uncomfortable (67.8%), with a higher proportion of high school-educated participants reporting this barrier than elementary- or college-educated participants (76.2% vs 64.1% and 67.8%, respectively, p=0.031).

### Motivators to participate in self-sample HPV testing

Over half of the 153 participants who reported returning the self-sampling kit (65.7% of respondents) found the self-sampling kit to be more convenient and less stressful compared to clinic-based cervical cancer screening (both 55.6%), with no significant differences between groups (*Table 3*). No patients found the self-sampling kit more embarrassing than the Pap test. While most participants found a Pap more embarrassing than the self-sampling kit (69.3%), significantly more Spanish- vs English-speaking participants found the Pap test more embarrassing than using a self-sampling kit (79.6% vs 53.3%, p=0.001). Participants with elementary or less education were more likely to report that a Pap was more embarrassing than high school- and college-educated participants (86.7% vs 65.2% and 55.0%, respectively, p=0.005).

Among participants who reported returning the HPV self-sampling kit, over half (59.5%) reported that the COVID-19 pandemic influenced their decision to participate in the HPV self-sampling trial (*Table 4*). The most commonly reported reasons for why the pandemic influenced the patient's decision to participate fell into three main categories: fear of getting COVID (41.3%), difficulty getting an appointment (21.7%), and having an easier time completing their screening at home (12%). Other reasons included not having time to travel, caring for children, and having a disability that made

**Table 1.** Participant characteristics among a subgroup of PRESTIS trial participants randomized to receive a mailed self-sample kit for HPV testing who participated in a telephone survey between August 2020 and September 2022 (n=233).

Participants were women, ages 30–65, who are patients in the Harris Health System (Harris County, TX) safety net healthcare system. Source data file: '*Table 1—source data 1*'.

| Patient characteristic | | M (SEM) | | |
|---|---|---|---|---|
| | | 47.2 (0.62) | | |
| **Age (years)** | | N (%) | | |
| | 30–39 | 59 (25.3%) | | |
| | 40–49 | 78 (33.5%) | | |
| | 50–59 | 69 (29.6%) | | |
| | 60–65 | 27 (11.6%) | | |
| | | N (%) | | |
| **Language of Interview** | English | 90 (38.6%) | | |
| | Spanish | 143 (61.4%) | | |
| | Hispanic | 162 (69.5%) | | |
| | Black/African American | 51 (21.9%) | | |
| | White | 14 (6.0%) | | |
| | Asian | 3 (1.3%) | | |
| **Race/Ethnicity** | Other | 3 (1.3%) | | |
| | Mexico | 92 (39.5%) | | |
| | United States | 81 (34.8%) | | |
| | Central America | 48 (20.6%) | | |
| | South America | 4 (1.7%) | | |
| | Asia | 2 (0.9%) | | |
| | Europe | 3 (1.3%) | | |
| | Other | 2 (0.9%) | | |
| **Place of birth** | Declined to answer | 1 (0.4%) | | |
| | | Total (n=233) | English (n=90) | Spanish (n=143) |
| | No formal schooling | 4 (1.7%) | 0 (0%) | 4 (2.8%) |
| | Some elementary | 15 (6.4%) | 0 (0%) | 15 (10.5%) |
| | Elementary | 45 (19.3%) | 3 (3.3%) | 42 (29.4%) |
| | Some high school | 41 (17.6%) | 13 (14.4%) | 28 (19.6%) |
| | High school | 64 (27.5%) | 28 (31.1%) | 36 (25.2%) |
| | Some college/vocational school | 33 (14.2%) | 21 (23.3%) | 12 (8.4%) |
| | College/vocational school | 28 (12.0%) | 25 (27.8%) | 3 (2.1%) |
| **Education completed*** | Declined to answer | 3 (1.3%) | 0 (0%) | 3 (2.1%) |

*Table 1 continued on next page*

*Table 1 continued*

| Patient characteristic | | M (SEM) | | |
|---|---|---|---|---|
| | <$10,000 | 27 (11.6%) | 16 (17.8%) | 11 (7.7%) |
| | $10,000 - $19,999 | 47 (20.2%) | 21 (23.3%) | 26 (18.2%) |
| | $20,000 - $29,999 | 29 (12.4%) | 13 (14.4%) | 16 (11.2%) |
| | $30,000 - $39,999 | 19 (8.2%) | 8 (8.9%) | 11 (7.7%) |
| | $40,000 - $49,999 | 8 (3.4%) | 4 (4.4%) | 4 (2.8%) |
| Household income | >$50,000 | 6 (2.6%) | 5 (5.6%) | 1 (0,7%) |
| | Declined to answer | 97 (41.6%) | 23 (25.6%) | 74 (51.7%) |

*p=0,000 Comparison of English- vs Spanish-speaking participants.

The online version of this article includes the following source data for table 1:

**Source data 1.** Telephone survey participant characteristics.

attending the clinic difficult. No significant differences in reported reasons were found between language groups.

## COVID-related barriers

Most participants who returned the kit (78.5%) reported that the COVID-19 pandemic affected their economic situation, 46.4% said it affected their mental health, and 39.2% said it affected their physical health (*Table 4*). Younger participants were more likely to report that the pandemic influenced their decision to participate (82.1% among 30–39, 51.9% among 40–49, 52.4% among 50–59, and 50% among 60 and older, p=0.010). Younger participants were also more likely to report that the pandemic had an economic effect on them than older participants (83.1% among 30–39, 83.4% among 40–49, 75.4% among 50–59, and 63% among 60 and older, p=0.015). More Spanish-speaking participants reported that COVID-19-related measures affected them economically (82.5%) compared to English-speaking participants (72.2%), though the results were not statistically significant (p=0.052). Conversely, significantly fewer Spanish-speaking participants reported that COVID-19 affected their mental health (37.8%) compared to English-speaking participants (60%, p=0.01). Participants with higher levels of education were more likely to report an effect on their mental health (42.2% among elementary or less, 44.8% among high school and 52.5% among college-educated participants, p=0.006). Most participants said the COVID-19 pandemic did not affect their physical health (60.5%), with participants with higher education levels more likely to report an effect on physical health (35.9% among elementary or less, 40% among high school and 41% among college-educated participants, p=0.033).

## Discussion

In our assessment of barriers to clinic-based screening during the COVID-19 pandemic, we found that discomfort with the test and with male providers, as well as embarrassment, are important and prevalent barriers to screening among under-screened safety net health system patients. These barriers were more prevalent among Hispanic women and those who completed the survey in Spanish. Our results suggest that barriers experienced by under-screened women within a safety net healthcare system may differ from those experienced by patients in other healthcare systems who have difficulty accessing care due to financial reasons and other barriers (*Fuzzell et al., 2021*; *Freeman, 2005*; *Akinlotan et al., 2017*). Similar to other studies conducted in safety net healthcare systems, we found that additional barriers beyond the access and financial barriers, including modesty concerns and discomfort, hinder participation in cervical cancer screening (*Fuzzell et al., 2021*; *Akinlotan et al., 2017*). One study conducted in low-income settings, with differing patient demographics and not limited to under-screened women, showed that cost was the most commonly-reported barrier (53.1%), and that anxiety (38.7%), embarrassment (25.6%), the anticipation of pain (23.6%), and being seen by a male physician (19.7%) were less important (*Akinlotan et al., 2017*). While this analysis reported similar

**Table 2.** Self-reported barriers among a subgroup of PRESTIS trial participants who were randomized to receive a mailed self-sample kit for HPV testing and completed a telephone survey from August 2020-September 2022 (n=233), Harris County, TX Source data file: '*Table 2—source data 1*'.

| | All (n=233) n (%) | Language | | | Race/Ethnicity | | | | | |
| --- | --- | --- | --- | --- | --- | --- | --- | --- | --- | --- |
| | | Spanish (n=143) n (%) | English (n=90) n (%) | p-value | Hispanic (n=162) n (%) | Black (n=51) n (%) | White (n=14) n (%) | Asian (n=3) n (%) | Other (n=3) n (%) | p-value |
| Getting a Pap is uncomfortable. Yes | 158 (67.8%) | 102 (71.3%) | 56 (62.2%) | 0.126 | 112 (69.1%) | 33 (64.7%) | 8 (57.1%) | 3 (100%) | 2 (33.3%) | 0.785 |
| No | 74 (31.8%) | 41 (28.7%) | 33 (36.7%) | | 49 (30.3%) | 18 (35.3%) | 6 (42.9%) | 0 (0%) | 1 (66.7%) | |
| Unsure | 1 (0.4%) | 0 (0%) | 1 (1.1%) | | 1 (0.6%) | 0 (0%) | 0 (0%) | 0 (0%) | 0 (0%) | |
| Uncomfortable with male provider Yes | 147 (63.1%) | 100 (69.9%) | 47 (52.2%) | 0.006 | 109 (67.3%) | 26 (51.0%) | 6 (42.9%) | 3 (100%) | 3 (100%) | 0.034 |
| No | 84 (36.1%) | 41 (28.7%) | 43 (47.8%) | | 52 (32.1%) | 25 (49.0%) | 7 (50.0%) | 0 (0%) | 0 (0%) | |
| Unsure | 2 (0.9%) | 2 (1.4%) | 0 (0%) | | 1 (0.6%) | 0 (0%) | 1 (7.1%) | 0 (0%) | 0 (0%) | |
| Getting a Pap is embarrassing Yes | 122 (52.4%) | 95 (66.4%) | 27 (30%) | 0.000 | 100 (61.7%) | 13 (25.5%) | 6 (42.9%) | 2 (66.7%) | 1 (33.3%) | 0.000 |
| No | 109 (46.8%) | 48 (33.6%) | 61 (67.8%) | | 61 (37.7%) | 37 (72.6%) | 8 (57.1%) | 0 (0%) | 2 (66.7%) | |
| Unsure | 2 (0.9%) | 0 (0%) | 2 (2.2%) | | 1 (0.6%) | 1 (2.0%) | 0 (0%) | 1 (33.3%) | 0 (0%) | |
| Getting a Pap test is expensive * Yes | 47 (20.3%) | 36 (25.6%) | 11 (12.2%) | 0.024 | 39 (24.2%) | 6 (11.8%) | 0 (0%) | 1 (66.7%) | 1 (33.3%) | 0.049 |
| No | 159 (68.5%) | 94 (66.2%) | 65 (72.2%) | | 104 (64.6%) | 38 (74.5%) | 0 (0%) | 2 (33.3%) | 1 (33.3%) | |
| Unsure | 26 (11.2%) | 12 (8.5%) | 14 (15.6%) | | 18 (11.2%) | 7 (13.7%) | 14 (100%) | 0 (0%) | 1(33.3%) | |

| | Age | | | | | Education† | | | |
| --- | --- | --- | --- | --- | --- | --- | --- | --- | --- |
| | 30–39 (n=59) | 40–49 (n=78) | 50–59 (n=69) | 60–65 (n=27) | p-value | Elementary or less (n=64) | Some/all High school (n=105) | Some/all College (n=61) | p-value |
| Getting a Pap is uncomfortable. Yes | 40 (67.8%) | 57 (73.1%) | 40 (58.0%) | 21 (77.8%) | 0.160 | 41 (64.1%) | 80 (76.2%) | 35 (67.8%) | 0.031 |
| No | 18 (30.5%) | 21 (26.9%) | 29 (42.0%) | 6 (22.2%) | | 23 (35.9%) | 24 (22.9%) | 26 (42.6%) | |
| Unsure | 1 (1.7%) | 0 (0%) | 0 (0%) | 0 (0%) | | 0 (0.0%) | 1 (1.0%) | 0 (0%) | |
| Uncomfortable with male provider Yes | 36 (61.0%) | 46 (59.0%) | 46 (66.7%) | 19 (70.4%) | 0.357 | 45 (70.3%) | 66 (62.9%) | 34 (55.7%) | 0.298 |
| No | 22 (37.3%) | 32 (41.0%) | 23 (33.3%) | 7 (25.9%) | | 18 (28.1%) | 38 (36.2%) | 27 (44.3%) | |
| Unsure | 1 (1.7%) | 0 (0%) | 0 (0%) | 1 (3.7%) | | 1 (1.6%) | 1 (1.0%) | 0 (0%) | |
| Getting a Pap is embarrassing Yes | 29 (49.2%) | 43 (55.1%) | 31 (44.9%) | 19 (70.4%) | 0.252 | 36 (56.3%) | 62 (59.1%) | 22 (36.1%) | 0.021 |
| No | 30 (50.9%) | 34 (43.6%) | 37 (53.6%) | 8 (29.6%) | | 28 (43.8%) | 42 (40.0%) | 38 (62.3%) | |
| Unsure | 0 (0%) | 1 (1.3%) | 1 (1.5%) | 0 (0%) | | 0 (0%) | 1 (1.0%) | 1 (1.6%) | |
| Getting a Pap test is expensive * Yes | 15 (25.4%) | 12 (15.4%) | 14 (203%) | 6 (23.1%) | 0.842 | 17 (27.0%) | 24 (22.9%) | 6 (9.8%) | 0.12 |
| No | 38 (64.4%) | 58 (74.4%) | 46 (66.7%) | 17 (65.4%) | | 44 (69.8%) | 65 (61.9%) | 47 (77.1%) | |
| Unsure | 6 (10.2%) | 8 (10.3%) | 9 (13.0%) | 3 (11.5%) | | 2 (3.2%) | 16 (15.2%) | 8 (13.1%) | |

*Missing n=1.
†Missing = 3.

The online version of this article includes the following source data for table 2:

**Source data 1.** Telephone survey results of barriers to getting provider-performed clinic-based screening among underscreened women participating in the PRESTIS study.

**Table 3.** Motivators- Self-sampling vs Pap among participants who self-reported completing the at-home self-sampling kit for HPV testing during a telephone survey (n=153), August 2020-September 2022, Harris County, TX Source data file: '*Table 3—source data 1*'.

| | All (N=153) n (%) | Language | | | Race/Ethnicity | | | | | |
| --- | --- | --- | --- | --- | --- | --- | --- | --- | --- | --- |
| | | Spanish N=93 n (%) | English N=60 n (%) | p-value | Hispanic (n=108) n (%) | Black (n=33) n (%) | White (n=7) n (%) | Asian (n=2) n (%) | Other (n=3) n (%) | p-value |
| Convenience of Pap vs self-sampling Self-sampling more convenient Pap more convenient Both are about the same | 85 (55.6%) 18 (11.8%) 50 (32.7%) | 48 (51.6%) 14 (15.1%) 31 (33.3%) | 37 (61.7%) 19 (31.7%) 4 (6.7%) | 0.237 | 58 (53.7%) 14 (13.0%) 36 (33.3%) | 19 (57.6%) 3 (9.1%) 11 (33.3%) | 6 (85.7%) 0 (0%) 1 (14.3%) | 1 (50.0%) 0 (0%) 1 (50%) | 1 (33.3%) 1 (33.3%) 1 (33.3%) | 0.756 |
| Embarrassment of Pap vs self-sampling Self-sampling more embarrassing Pap more embarrassing Both are about the same | 0 (0%) 106 (69.3%) 47 (30.7%) | 0 (0%) 74 (79.6%) 19 (20.4%) | 0 (0%) 32 (53.3%) 28 (46.7%) | 0.001 | 0 (0%) 79 (73.2%) 29 (26.9%) | 0 (0%) 18 (54.6%) 15 (45.5%) | 0 (0%) 4 (57.1%) 3 (42.9%) | 0 (0%) 2 (100%) 0 (0%) | 0 (0%) 3 (100%) 0 (0%) | 0.158 |
| Stress/anxiety of Pap vs self-sampling Self-sampling more stressful Pap more stressful Both are about the same | 6 (3.9%) 85 (55.6%) 62 (40.5%) | 4 (4.3%) 52 (55.9%) 37 (39.8%) | 2 (3.3%) 33 (55%) 25 (41.7%) | 0.940 | 6 (5.6%) 60 (55.6%) 42 (38.9%) | 0 (0%) 18 (54.6%) 15 (45.5%) | 0 (0%) 4 (57.1%) 3 (42.9%) | 0 (0%) 0 (0%) 2 (100%) | 0 (0%) 3 (100%) 0 (0%) | 0.479 |

**Age**

**Education\*\***

| | Age | | | | | Education\*\* | | | |
| --- | --- | --- | --- | --- | --- | --- | --- | --- | --- |
| | 30–39 (n=39) | 40–49 (n=52) | 50–59 (n=42) | 60–65 (n=20) | p-value | Elementary or less (n=45) | Some/all High school (n=66) | Some/all College (n=40) | p-value |
| Convenience of Pap vs self-sampling Self-sampling more convenient Pap more convenient Both are about the same | 23 (59.0%) 1 (2.6%) 15 (38.5%) | 29 (55.8%) 9 (17.3%) 14 (26.9%) | 20 (47.6%) 8 (19.1%) 14 (33.3%) | 13 (65.0%) 0 (0%) 7 (35.0%) | 0.095 | 22 (48.9%) 7 (15.6%) 16 (35.6%) | 37 (56.1%) 7 (10.6%) 22 (33.3%) | 24 (60.05) 4 (10.0%) 12 (30.0%) | 0.841 |
| Embarrassment of Pap vs self-sampling Self-sampling more embarrassing Pap more embarrassing Both are about the same | 0 (0%) 25 (64.1%) 14 (35.9%) | 0 (0%) 38 (73.1%) 14 (26.9%) | 0 (0%) 26 (61.9%) 16 (38.1%) | 0 (0%) 17 (85.0%) 3 (15.0%) | 0.242 | 0 (0%) 39 (86.7%)6 (13.3%) | 0 (0%) 43 (65.2%) 23 (34.9%) | 0 (0%) 22 (55.0%) 18 (45.0%) | 0.005 |
| Stress/anxiety of Pap vs self-sampling Self-sampling more stressful Pap more stressful Both are about the same | 2 (5.1%) 23 (59.0%) 14 (35.9%) | 1 (1.9%) 29 (55.8%) 22 (42.3%) | 3 (7.1%) 19 (45.2%) 20 (47.6%) | 0 (0%) 14 (70.0%) 6 (30.0%) | 0.529 | 0 (0%) 27 (60.0%) 18 (40.0%) | 5 (7.6%) 36 (54.6%) 25 (37.9%) | 1 (2.5%) 21 (52.5%) 18 (45.0%) | 0.318 |

The online version of this article includes the following source data for table 3:

**Source data 1.** Telephone survey results showing motivators of self-sampling vs. Pap.

**Table 4.** COVID-related barriers self-reported by PRESTIS trial participants who completed a telephone survey between August 2020 and September 2022, Harris County, TX Source data file: '*Table 4—source data 1*'.

| | | Language | | | Race/Ethnicity | | | | | |
| | All N=233 n (%) | Spanish N=143 n (%) | English N=90 n (%) | p-value | Hispanic (n=162) n (%) | Black (n=51) n (%) | White (n=14) n (%) | Asian (n=3) n (%) | Other (n=3) n (%) | p-value |
|---|---|---|---|---|---|---|---|---|---|---|
| **COVID barriers for all patients (n=233)** | | | | | | | | | | |
| COVID-19 had economic effect Yes- large amount | 101 (43.4%) | 60 (42%) | 41 (45.6%) | | 68 (42.0%) | 24 (47.1%) | 4 (28.6%) | 3 (100%) | 2 (66.6%) | |
| Yes- small amount | 82 (35.2%) | 58 (40.6%) | 24 (26.7%) | | 62 (38.35) | 16 (31.4%) | 4 (28.6%) | 0 (0%) | 0 (0%) | |
| No | 50 (21.5%) | 25 (17.5%) | 25 (17.5%) | 0.052 | 32 (19.8%) | 11 (21.6%) | 6 (42.9%) | 0 (0%) | 1 (33.3%) | 0.280 |
| COVID-19 affected mental health Yes- large amount | 34 (14.6%) | 13 (9.1%) | 21 (23.3%) | | 17 (10.5%) | 14 (27.5%) | 2 (14.3%) | 0 (0%) | 1 (33.3%) | |
| Yes- small amount | 74 (31.8%) | 41 (28.7%) | 33 (36.7%) | | 50 (30.9%) | 19 (37.3%) | 3 (21.4%) | 1 (33.3%) | 1 (33.3%) | |
| No | 125 (53.7%) | 89 (62.2%) | 36 (40%) | 0.001 | 95 (58.6%) | 18 (35.3%) | 9 (64.3%) | 2 (66.7%) | 1 (33.3%) | 0.034 |
| COVID-19 affected physical health Yes- large amount | 35 (15%) | 20 (14%) | 15 (16.7%) | | 20 (12.4%) | 11 (25.6%) | 3 (21.4%) | 0 (0)% | 1 (33.3%) | |
| Yes- small amount | 57 (24.5%) | 33 (23.1%) | 24 (26.7%) | | 42 (25.9%) | 12 (23.5%) | 1 (7.1%) | 1 (33.3%) | 1 (33.3%) | |
| No | 141 (60.5%) | 90 (62.9%) | 51 (56.7%) | 0.633 | 100 (61.7%) | 28 (54.9%) | 10 (71.4%) | 2 (66.7%) | 1 (33.3%) | 0.382 |

| Age | | | | | Education† | | | |
|---|---|---|---|---|---|---|---|---|
| 30–39 (n=59) | 40–49 (n=78) | 50–59 (n=69) | 60–65 (n=27) | p-value | Elementary or Less (n=64) | Some/all High School (n=105) | Some/all College (n=40) | p-value |

*Table 4 continued on next page*

**COVID barriers for all patients (n=233)**

| | All n (%) | Language | | | Age | | | | |
|---|---|---|---|---|---|---|---|---|---|
| | | Spanish n (%) | English n (%) | p-value | 30–39 n (%) | 40–49 n (%) | 50–59 n (%) | 60–65 n (%) | p-value |
| **COVID-19 had economic effect** | | | | 0.015 | | | | | 0.364 |
| Yes- large amount | 19 (32.2%) | 36 (46.2%) | 36 (52.2%) | | 10 (37.0%) | 21 (32.8%) | 48 (45.7%) | 30 (49.2%) | |
| Yes- small amount | 30 (50.9%) | 29 (37.2%) | 16 (23.2%) | | 7 (25.9%) | 28 (43.8%) | 34 (32.4%) | 19 (31.2%) | |
| No | 10 (17.0%) | 13 (16.7%) | 17 (24.6%) | | 10 (37.0%) | 15 (23.4%) | 23 (21.9%) | 12 (19.7%) | |
| **COVID-19 affected mental health** | | | | 0.947 | | | | | 0.006 |
| Yes- large amount | 7 (11.9%) | 11 (14.1%) | 11 (15.9%) | | 5 (18.5%) | 3 (4.7%) | 23 (21.9%) | 7 (11.5%) | |
| Yes-small amount | 21 (35.6%) | 23 (29.5%) | 23 (33.3%) | | 7 (25.9%) | 24 (37.5%) | 24 (22.9%) | 25 (41.0%) | |
| No | 31 (52.5%) | 44 (56.4%) | 35 (50.7%) | | 15 (55.6%) | 37 (57.8%) | 58 (55.2%) | 29 (47.5%) | |
| **COVID-19 affected physical health** | | | | 0.762 | | | | | 0.033 |
| Yes- large amount | 7 (11.9%) | 13 (16.7%) | 12 (17.4%) | | 3 (11.1%) | 4 (6.3%) | 23 (21.9%) | 7 (11.5%) | |
| Yes- small amount | 12 (20.3%) | 19 (24.4%) | 20 (29.0%) | | 6 (22.2%) | 19 (29.7%) | 19 (18.1%) | 18 (29.5%) | |
| No | 40 (67.8%) | 46 (59.0%) | 37 (53.6%) | | 18 (66.7%) | 41 (64.1%) | 63 (60.0%) | 36 (59.0%) | |

**COVID barriers among those who completed self-sample kit (n=153)***

| | All (N=153) n (%) | Language | | | Age | | | | |
|---|---|---|---|---|---|---|---|---|---|
| | | Spanish N=93 n (%) | English N=60 n (%) | p-value | 30–39 (n=39) n (%) | 40–49 (n=52) n (%) | 50–59 (n=69) n (%) | 60–65 (n=27) n (%) | p-value |
| **COVID-19 affected participation in HPV self-sampling trial** | | | | 0.530 | | | | | 0.010 |
| Yes | 91 (59.5%) | 58 (62.4%) | 33 (55%) | | 32 (82.1%) | 27 (51.9%) | 22 (52.4%) | 10 (50%) | |
| No | 61 (39.9%) | 34 (36.6%) | 27 (45%) | | 7 (18.0%) | 25 (48.1%) | 19 (45.3%) | 10 (50%) | |
| Don't know | 1 (0.7%) | 1 (1.1%) | 0 (0%) | | 0 (0%) | 0 (0%) | 1 (2.4%) | 0 (0%) | |

*Table 4 continued*

**COVID barriers for all patients (n=233)**

| | Categories of COVID barriers reported by those who reported that COVID affected their participation in the trial (n=92)* | | | |
|---|---|---|---|---|
| | All N=92 n (%) | Spanish N=59 n (%) | English N=33 n (%) | p-value |
| Way that COVID affected participation | 38 (41.3%) | 24 (40.7%) | 14 (42.4%) | |
| Fear of getting COVID | 20 (21.7%) | 16 (27.1%) | 4 (12.1%) | |
| Difficulties getting appointment | 11 (12%) | 7 (11.9%) | 4 (12.1%) | |
| Easier at home Other | 23 (25%) | 12 (20.3%) | 11 (33.3%) | 0.304 |

*No significant differences were found with race/ethnicity or education.

†Missing n=3.

The online version of this article includes the following source data for table 4:

**Source data 1.** Telephone results showing COVID-related barriers self-reported by participants.

barriers to this study, we found that less than a quarter of respondents reported that screening was expensive, and over half reported embarrassment, anxiety, and being seen by a male provider were barriers. This indicates the need to evaluate barriers within each environment to implement strategic programs to increase screening.

The motivators for using an at-home HPV self-sampling kit appear to address key barriers to cervical cancer screening found in our survey participants. Most who used the kit found it to be less stressful, embarrassing, and more convenient than clinic-based screening. Significantly more Spanish-speaking women and women with lower education completion found the at-home kits to be less embarrassing than clinic-based screening, a barrier reported significantly more among those groups of participants. Our findings suggest that self-sampling kits may address or circumvent some of the key barriers reported by survey participants within a safety net health system, especially those reported by Spanish-speaking women, and may help to address disparities in cervical cancer screening adherence. Our findings are consistent with other studies in showing that self-sampling can address barriers to clinic-based screening in various countries and health system settings (*Herrington, 2022*), though ours appears to be the first to report results from US safety net health system under-screened women.

Our results show that COVID-19 was a motivating factor for most respondents to participate in the at-home self-sampling HPV trial and that many patients experienced additional barriers to care since the beginning of the pandemic. The most common barriers included difficulty making appointments, fear of getting COVID, and a broad response that screening was easier at home. While the survey did not probe about this last response, many participants mentioned it in the context of competing priorities amid the pandemic, such as childcare. These responses align with research indicating that the burden of childcare and elder care has fallen disproportionately on women during the pandemic (*Byrd et al., 2007*).

This study had certain limitations that should be considered when interpreting the results. Because the study was conducted among women in a safety net system that cares for un- and under-insured individuals, results may not be generalizable to women served by other types of health systems. Similarly, the main analysis, while relevant to the safety net system in which the study was conducted, may not apply to other healthcare systems, or to international audiences. Women in the community and other healthcare settings often face significant structural barriers related to access to care due to lack of insurance and/or cost. The prevalent barriers in our study most certainly reflect that financial and insurance barriers are largely removed due to participants' enrollment in the health system. Additionally, as mentioned, the closed-ended survey format did not allow us to probe into some of the responses, particularly how the COVID-19 pandemic influenced the use of the kit. Nonetheless, this study is unique in that it gives in-depth insight into the particular barriers experienced by safety net patients during the COVID-19 pandemic. To our knowledge, this is the first analysis of cervical cancer screening barriers among under-screened women in a safety net healthcare system in the COVID-19 era.

In conclusion, mailed at-home HPV self-sampling kits present an opportunity to reduce important barriers to cervical cancer screening among women in a safety net healthcare system. Furthermore, during the COVID-19 pandemic, these barriers may have been exacerbated by the economic, physical, and mental effects of the pandemic. Further research is needed to understand additional barriers experienced by women during the COVID-19 pandemic and how these might be addressed with new screening tools such as at-home HPV testing using self-sampling. Implementation of new screening programs should address the specific barriers to clinic-based screening and motivators to self-sampling experienced by their patient populations.

## Acknowledgements

The authors would like to thank Harris Health System for their partnership and support throughout the study. This study is supported by a grant from the National Institute for Minority Health and Health Disparities (NIMHD, R01MD013715, PI: JR Montealegre). The NIMHD was not involved in the study design; the collection, analysis, or interpretation of data; the writing of this manuscript; or the decision to submit the manuscript for publication. The REDCap software platform used for data capture is supported by a grant from the National Center for Supporting Translational Sciences (UL1 TR000445).

## Additional information

### Funding

| Funder | Grant reference number | Author |
|--------|------------------------|--------|
| National Institute for Minority Health and Health Disparities | R01MD013715 | Jane R Montealegre |

The funders had no role in study design, data collection and interpretation, or the decision to submit the work for publication.

### Author contributions

Susan Parker, Conceptualization, Data curation, Formal analysis, Methodology, Writing – original draft, Project administration, Writing – review and editing; Ashish A Deshmukh, Conceptualization, Investigation, Writing – review and editing; Baojiang Chen, David R Lairson, Sally W Vernon, Supervision, Writing – review and editing; Maria Daheri, Project administration, Writing – review and editing; Jane R Montealegre, Conceptualization, Funding acquisition, Investigation, Methodology, Writing – review and editing

### Author ORCIDs

Susan Parker ⓘ http://orcid.org/0000-0002-6722-4717

### Ethics

Clinical trial registration NCT03898167.

Human subjects: The survey was administered by trained, bilingual researcher coordinators in the patient's preferred language (English or Spanish). Participants were asked to provide verbal consent before commencing the survey and were sent a $20 gift card upon completion. This research was reviewed and approved by Baylor College of Medicine and Harris Health System's Institutional Review Boards (IRBs) (protocol ID H-44944). A waiver of written documentation of informed consent was granted by the IRBs for participants in the parent study given the minimal risks involved and to minimize participation bias. An IRB-approved verbal consent script was read to participants randomized to participate in the telephone survey. After being read the script, patients were given the opportunity to ask questions, and verbally indicated whether they consented to participate in the telephone survey. A consent to publish was not obtained as only aggregate data will be reported.

### Decision letter and Author response

Decision letter https://doi.org/10.7554/eLife.84664.sa1
Author response https://doi.org/10.7554/eLife.84664.sa2

---

## Additional files

### Supplementary files

• MDAR checklist

### Data availability

All data generated or analyzed during this study are included in the manuscript and supporting file. Source data files have been provided for Tables 1–4.

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
