## [Editor Report]

The evidence presented in the manuscript is solid, and the study is a valuable contribution to research on at-home sampling for cervical cancer screening in underserved populations. The fact that the study was conducted during the COVID-19 pandemic makes it particularly informative for policymaking in circumstances of restricted access to care.

---

## [Decision Letter]

**Decision letter after peer review:**

Thank you for submitting your article "Perceived barriers to cervical cancer screening and motivators for at-home HPV self-sampling during the COVID-19 pandemic: A telephone survey of randomized controlled trial participants" for consideration by *eLife*. Your article has been reviewed by 4 peer reviewers, and the evaluation has been overseen by a Reviewing Editor and a Senior Editor. The following individuals involved in the review of your submission have agreed to reveal their identity: Matejka Rebolj (Reviewer #3); Paolo Giorgi Rossi (Reviewer #4).

As is customary in *eLife*, the reviewers have discussed their critiques with one another. What follows below is the Reviewing Editor's edited compilation of the essential and ancillary points provided by reviewers in their critiques and in their interaction post-review. Please submit a revised version that addresses these concerns directly. Although we expect that you will address these comments in your response letter, we also need to see the corresponding revision clearly marked in the text of the manuscript. Some of the reviewers' comments may seem to be simple queries or challenges that do not prompt revisions to the text. Please keep in mind, however, that readers may have the same perspective as the reviewers. Therefore, it is essential that you attempt to amend or expand the text to clarify the narrative accordingly.

Essential revisions:

1. The manuscript mainly focuses on differences between English-speaking and Spanish-speaking participants, presented in Tables 2 to 4, but comparisons by age, screening history, income, etc. may also be interesting (reviewers 1,3, and 4). Please include extra comparisons by age, income, etc., or convincingly argue why the focus is only on different language groups. For the additional analyses, a multivariable approach may be chosen in which different determinants are compared, as suggested by reviewer 1.

2. The manuscript contains many inaccuracies and inconsistencies. Most of them have been indicated by the reviewers. Please follow their suggestions as closely as possible. In addition, make sure that the table numbers match with the order in which they appear in the manuscript.

3. The reviewers had questions about missing items/data (reviewers 1 and 2) and inclusion/exclusion criteria (reviewers 3 and 4). Please make sure that the information is complete.

4. Table headings should provide more detailed information about the participants so that tables can be read without consulting the main text.

5. Do not use statements like p <.05, but give precise p-values, as noted by reviewer 2. Also, indicate in the Material and Methods section when Fisher's exact test is used and when the chi-square test is used.

6. The manuscript should provide a better overview of the existing literature. Many studies have investigated the attitude of underscreened women from high-income countries towards self-sampling, as noted by reviewer 3. Note that the recently published IARC Handbook on Cervical Cancer Screening (Volume 18) contains a lot of references to HPV self-sampling.

*Reviewer #1 (Recommendations for the authors):*

Susan L. Parker et al. interviewed 233 women (being a subgroup of the PRESTIS study) from a US safety-net system on their perceived barriers and motivators for at-home HPV self-sampling during the COVID-19 pandemic. Five patient characteristics were given of which language was used in a monovariate analysis to discriminate the results of (1) the self-reported barriers of getting a Pap smear, (2) motivators for self-sampling versus Pap, and (3) COVID-related barriers.

The differences observed in the barriers to having clinician-taken Pap smears between different health care systems will inform health professionals and policymakers to stake at the mutual, universal ones or act specifically on barriers for a particular group, hence creating a maximal impact on cervical cancer screening participation. In this study, Spanish-speaking women stressed the embarrassment of having a Pap (by a male provider, compared to the self-sample) more than their English-speaking counterparts within the same safety net system. The survey results on the motivators to use an HPV self-sampling kit confirm previous and worldwide experiences, putting self-sampling as a convenient intervention to address under-screened women, also in this particular setting. The COVID-pandemic impacted as well on the willingness to accept self-sampling as a valuable alternative.

Strengths:

The manuscript is written straightforwardly, and highly accessible to the general public, with a clear-cut introduction on the impact of COVID on primary care, focusing on cervical cancer screening in an under-served population. Results engage on the differences between Spanish- (2/3 of the subgroup) and English-speaking women (1/3) from the same safety net system, probably revealing cultural differences between them (although not discussed).

Weaknesses:

Authors have to be careful in their word choice, starting in the abstract and throughout the manuscript:

– Home-based HPV testing, might give the reader the impression that women not only took the sample themselves but also obtained the result of the test at the same time, as with a pregnancy test. This is not the case for HPV testing which is up-to-now always performed in a laboratory. Better to formulate as home-based self-sampling for HPV testing.

– Authors should provide absolute numbers (n/N) next to percentages of the perceived barriers/motivators, instead of using non-scientific scorings like 'over half' and 'most' (for percentages ranging from 52.4% up to 69.3%?). 'Most' implies almost all.

– Are there other options for women besides going to a clinic to obtain the classic Pap smear? If so, rather use 'clinician-based' screening or sampling instead of 'clinic-based'.

The total overview of the PRESTIS study and the subgroup being part of the telephone survey is only clear from the CONSORT Flow Diagram, but the specific participants focusing on in this manuscript should be described in detail to inform the reader of the magnitude and representativeness of the survey group. Even from the Flow Diagram, it is not clear which number of respondents returned the self-sampling kit (presumably 153) and which part did not (N=80?).

The methods section describes the questions used in the questionnaire, but items like 'telehealth access' cannot be found in the tables or Results section. Moreover, the self-sampling kit is not detailed (vaginal brush/urine sample/…? Which specific device? Which procedure?).

The patient characteristics (demographic data, SES) presented in Table 1 are very limited and specifically 'age' and 'screening history' are lacking (besides 'employment' and 'marital status'), which are important risk factors for studying cervical cancer screening. These data could however easily be obtained during the phone calls, or might even be available from the overall PRESTIS study. Please add if available. These variables would have been very important in bivariate or multivariate analysis to determine the variables associated with HPV self-sampling uptake and the barrier outcomes, to enrich the basic descriptive statistic presented here. Furthermore, the denominator of Table 1 is indicated to be 233, but some variables have missing data, which are not indicated in the table, nor discussed. Specifically, the variable 'household income', being a very important SES factor, was only reported for 136 women. Please report all missing data, and how this could have influenced the distribution shown.

Table 4 should be structured better to make clear to the reader that question 2 only relates to the positive answers in question 1. The denominator shown in the title of this table (N=153) only applies to the first question, the last 3 show data for all 233 interviewed women. Please adjust accordingly.

A more thorough discussion on the overlapping barriers reported in different healthcare systems and the specific deviating barriers – and the reasons why they deviate – in this specific patient group, would have been very informative.

Reviewer #1 (Recommendations for the authors):

Please find most recommendations in the public review, complete with:

(Abstract)

– … reported clinic(clinician)-based screening… discomfort seeing male providers.

Might be better formulated as: … is experiencing discomfort when seeing male providers.

– Do you 'complete' the kit or just 'use' it?

(Materials and methods)

– Can be clarified more.

– Leave the verbs away in the details on the inclusion criteria for the PRESTIS study.

(Results)

– Given results can be summarized and condensed, and should be enlarged with more in-depth analysis if the data are available.

– The percentage of the Mexico-born population should be corrected (39.4% in the text versus 39.7% in Table 1).

(Tables)

– Table 1: Give correct denominators for each variable.

– Table 2: give N for both Spanish- and English-speaking subgroups.

– Table 4: restructure and give the correct denominator for each question.

*Reviewer #2 (Recommendations for the authors):*

This manuscript provides important information on barriers to cervical cancer screening participation, and the potential of self-sampling kits to increase participation, especially among ethnic and/or linguistic minority women.

The manuscript is clearly and concisely written. The tables lack important information (esp. age of the participating women). P-values in the text are not given as precise values but whether they are below some cutoff; this is not recommended. The flow of the manuscript can be improved. One of the conclusions ("Furthermore, during the COVID-19 pandemic, these barriers were exacerbated by economic, physical, and mental effects of the pandemic") is not substantiated and could be rephrased.

The main conclusion in the Discussion "In conclusion, mailed at-home HPV self-sampling kits present an opportunity to reduce important barriers to cervical cancer screening among women in a safety net healthcare system" stands and is an important message to cervical cancer screening organizations – there is a lot to gain with self-sampling kits.

p1, Abstract, and p7 and p8 Results: It is recommended to provide precise P values rather than stating P<0.01 or P<0.05, for all P>0. 0.001.

p7, Results: "Significantly more Spanish-speaking participants reported that getting a Pap test is embarrassing compared to non-Hispanic and English-speaking participants, 66.4% vs. 30%, p<0.01" The comparison reported here is correct for "English speaking participants", but was not shown in the analysis for non-Hispanics.

p7, Results: The order of reporting percentages and associations in the 2nd paragraph of Results is different from the order of reporting in Table 2. It is recommended to stick to the same order in the Table and text.

p7, Results: be consistent in the number of decimals given for percentages; throughout the Results, one decimal is provided, but in the 5th line from the bottom no decimal is given in "78%".

p7-8, Results, third paragraph: please refer to the appropriate Table here (Table 4).

p8, Results, second paragraph: I'd suggest that the authors first report the number of participants returning the kit before they report analyses restricted to that group, e.g. "153 (xx%) of participants returned the kit."

Results: it is recommended to number the Tables in order of appearance. Now Table 3 is first mentioned after Table 4 is mentioned.

Tables: it is recommended to provide full Table titles, stating Person, Place, and Time, so that a Table can be more easily understood on its own.

Table 1: It appears some data are missing (e.g. for Place of birth, Education, Household income). State in the body of the table or in a footnote which number of records had missing values for each variable.

Table 1 does not list age; age is crucial for any epidemiological analysis. Please add.

Table 2: the order of categories seems somewhat contra-intuitive: with the variable "Uncomfortable with male provider" the first category given is Yes, and the second No. Reverse order?

Same for the other variables in this Table 2?

Tables 2 and 3 and 4: Please provide the N above each of the three columns.

Table 2: Were P values calculates based on Yes and NO categories only, or also including the Unsure categories?

It appears "Getting a Pap test is expensive" has a missing value. Please note this in the body of the table or footnote.

Table 3: Indicate here in the title who is in the analysis, so as to explain there are only 153 in this Table rather than the 233 participants.

Discussion, p14: "Furthermore, during the COVID-19 pandemic, these barriers were exacerbated by economic, physical, and mental effects of the pandemic". I am not sure this has been demonstrated by the analyses the authors report.

References: Please check the rendering of ref 17.

*Reviewer #3 (Recommendations for the authors):*

The authors investigated attitudes toward HPV self-sampling as an alternative screening method for under-screened women who live with multiple social disadvantages in an urban area of the USA. Although their study included elements that were specific to the era of the COVID pandemic during which the study was undertaken, it once more confirmed that HPV self-sampling may be an attractive alternative for a proportion of under-screened women. Whether the positive attitudes described in the study would translate into increased screening participation, remains to be seen.

The authors could provide some more information on how women can seek screening from this health care provider. Does this provider offer cervical cancer screening, and did Arm 1 of the trial (control) actually represent "usual care"? If care is provided regardless of an individual's ability to pay, then why did some women complain that paps are too expensive? Do women need to seek screening themselves, or will they be reminded that they are (over)due? Around 2/3 of the women reported that they are uncomfortable around a male screening provider – how likely is it that a woman will be screened by a male vs. a female provider, and can she choose between the smear-taker's gender?

A question to clarify some of the exclusion criteria: why did women need to have had at least 2 visits with this health care provider in the last 3.5 years? Would this not tend to increase the likelihood that the study would not reach severely underscreened and never-screened women (e.g., those that do not engage with health care but could be motivated to do self-sampling)?

Why was the main analysis looking at English vs. Spanish-speaking participants? It may make sense for policy-making within the specific healthcare provider, but less so for the international audience. Could the authors report some other comparisons e.g., by age or certain other sociodemographic characteristics or the women's screening history? Also, from the exclusion criteria it seems that some women may have been screened only by paps. Did the attitudes towards HPV self-sampling among those women differ from the attitudes among women who were previously screened with co-testing (and indeed from the attitudes among unscreened women, if there were any in the study)? (A simple frequency distribution for such characteristics added to Table 1 would also be helpful)

The authors concluded that self-sampling "may help to address disparities in cervical cancer screening adherence". This is an abstract statement but could be better supported by evidence from their study, for example by reporting how many of the women approached for the trial actually did self-sampling. We know that in theory self-sampling should be able to increase screening participation, but translation into practice has not been extremely successful. That is why statements like these would better not be made unless they can be supported with data.

This analysis of barriers to clinician collection and attitudes around self-sampling comes after more than a decade of intensive research and multiple systematic reviews. The paper should provide a better overview of the existing literature. Several studies have investigated attitudes towards self-sampling and self-sampling uptake in disadvantaged women from high-income countries, which would be consistent with the population in this study, so relevant for comparison.

Finally, the authors stated that their study is the first to describe barriers in obtaining cervical cancer screening during the COVID-19 pandemic (for a specific population). This may be correct, but for most people, whether one likes it or not, life has moved on a while ago. Could the authors find a way to make this paper more interesting (and informative) for a post-pandemic world?

*Reviewer #4 (Recommendations for the authors):*

The paper affords an interesting issue and the whole project, including the trial, will timely assess the effectiveness of self-sampling in reducing inequalities in cervical cancer access.

It is now time to start implementing self-sampling and try to exploit the potential benefits of this device in increasing HPV test coverage. To do that we need such research because implementation must be context specific. In fact, in this field context-specific, implementation research is needed and results observed in trials conducted in other countries or in different health systems cannot be transferred to a different situation. Furthermore, the acceptability of self-sampling depends on the social and cultural background of the women, but also on how it is offered by the provider and on how the communication between the provider and women occurs.

This study will integrate data from a survey and from a trial. In this paper, the authors report the results of the survey. It highlights that part of the barriers reducing access to screening of disadvantaged women can really be removed by self-sampling. The study shows that these barriers are now more important for Hispanic women. These findings are important, but more information from this survey can be exploited.

Methods

Data collection

I did not understand if all trial participants have been contacted for the interview. Furthermore, I did not understand if the consent was only to participate in the survey or in the trial (it would be a strong limitation to the trial including only those agreeing to participate, even if sometimes Ethics Committee or IRB imposes it).

I think it should be reported a description of the survey power or the sample size determination, whether it has been established on the basis of a formal sample size estimation or with a convenience constraint.

Measures

It is not clear if the different concepts (embarrassment, discomfort…) are measured in a single question or in separate items.

Results

It is important to report the response rate to the survey. It would be important also to report if the response was differential among those who returned the self-sample (or attended) and those who did not.

Why did the authors investigate only ethnicity as a possible determinant? Not the age, income, or education? Returning the kit or not? I think such analyses would be relevant to your objectives.

Discussion

It is balanced and well-written.

I suggest discussing the limitations in light of other results, not in a separate paragraph.

---

## [Author Response]

Essential revisions:1. The manuscript mainly focuses on differences between English-speaking and Spanish-speaking participants, presented in Tables 2 to 4, but comparisons by age, screening history, income, etc. may also be interesting (reviewers 1,3, and 4). Please include extra comparisons by age, income, etc., or convincingly argue why the focus is only on different language groups. For the additional analyses, a multivariable approach may be chosen in which different determinants are compared, as suggested by reviewer 1.

Thank you for this feedback. These additional comparisons will be made using the parent trial results (n=2,268), as well as an upcoming acceptability paper using the telephone survey results. We have stratified Table 1 to show significant differences in education level between the language groups (page 10). We have now also included additional analyses in Tables 2-4 that show barriers and motivators by race/ethnicity, age and education level. Income was not included in these tables because of the relatively large number of missing data for this variable.

2. The manuscript contains many inaccuracies and inconsistencies. Most of them have been indicated by the reviewers. Please follow their suggestions as closely as possible. In addition, make sure that the table numbers match with the order in which they appear in the manuscript.

Responses to individual reviewer comments and suggestions have been indicated below in blue and changes have been made within the manuscript to address each suggestion.

3. The reviewers had questions about missing items/data (reviewers 1 and 2) and inclusion/exclusion criteria (reviewers 3 and 4). Please make sure that the information is complete.

This has been addressed please see below.

4. Table headings should provide more detailed information about the participants so that tables can be read without consulting the main text.

This has been addressed in the updated manuscript file (pages 9-12).

5. Do not use statements like p <.05, but give precise p-values, as noted by reviewer 2. Also, indicate in the Material and Methods section when Fisher's exact test is used and when the chi-square test is used.

Exact p-values have been added (pages 1, 7-9) and a statement about the use of Fisher’s exact test has been added (page 6).

6. The manuscript should provide a better overview of the existing literature. Many studies have investigated the attitude of underscreened women from high-income countries towards self-sampling, as noted by reviewer 3. Note that the recently published IARC Handbook on Cervical Cancer Screening (Volume 18) contains a lot of references to HPV self-sampling.

Thank you for this suggestion. We have included a more thorough discussion of the literature on acceptability and attitudes toward self-sampling. These include citations from the IARC Handbook in the introduction, as well as additional literature from other global settings.

Reviewer #1 (Recommendations for the authors):Susan L. Parker et al. interviewed 233 women (being a subgroup of the PRESTIS study) from a US safety-net system on their perceived barriers and motivators for at-home HPV self-sampling during the COVID-19 pandemic. Five patient characteristics were given of which language was used in a monovariate analysis to discriminate the results of (1) the self-reported barriers of getting a Pap smear, (2) motivators for self-sampling versus Pap, and (3) COVID-related barriers.The differences observed in the barriers to having clinician-taken Pap smears between different health care systems will inform health professionals and policymakers to stake at the mutual, universal ones or act specifically on barriers for a particular group, hence creating a maximal impact on cervical cancer screening participation. In this study, Spanish-speaking women stressed the embarrassment of having a Pap (by a male provider, compared to the self-sample) more than their English-speaking counterparts within the same safety net system. The survey results on the motivators to use an HPV self-sampling kit confirm previous and worldwide experiences, putting self-sampling as a convenient intervention to address under-screened women, also in this particular setting. The COVID-pandemic impacted as well on the willingness to accept self-sampling as a valuable alternative.Strengths:The manuscript is written straightforwardly, and highly accessible to the general public, with a clear-cut introduction on the impact of COVID on primary care, focusing on cervical cancer screening in an under-served population. Results engage on the differences between Spanish- (2/3 of the subgroup) and English-speaking women (1/3) from the same safety net system, probably revealing cultural differences between them (although not discussed).Weaknesses:Authors have to be careful in their word choice, starting in the abstract and throughout the manuscript:– Home-based HPV testing, might give the reader the impression that women not only took the sample themselves but also obtained the result of the test at the same time, as with a pregnancy test. This is not the case for HPV testing which is up-to-now always performed in a laboratory. Better to formulate as home-based self-sampling for HPV testing.

Thank you for pointing out the potentially misleading terminology. We have updated the terminology to “home-based self-sample HPV testing” or “HPV self-sampling” and do not refer to the procedure as “HPV testing” (pages 1, 3-4).

– Authors should provide absolute numbers (n/N) next to percentages of the perceived barriers/motivators, instead of using non-scientific scorings like 'over half' and 'most' (for percentages ranging from 52.4% up to 69.3%?). 'Most' implies almost all.

Thank you for this feedback. We have updated the results to include percentages and tempered the language so as to not overstate the percentages. Absolute numbers can be found in Tables 1-4.

– Are there other options for women besides going to a clinic to obtain the classic Pap smear? If so, rather use 'clinician-based' screening or sampling instead of 'clinic-based'.

In this study, women were only provided with the options of going to a clinic for a standard Pap/HPV co-test or receiving an at-home self-sampling kit for HPV testing. Thus, no changes were made to this terminology.

The total overview of the PRESTIS study and the subgroup being part of the telephone survey is only clear from the CONSORT Flow Diagram, but the specific participants focusing on in this manuscript should be described in detail to inform the reader of the magnitude and representativeness of the survey group. Even from the Flow Diagram, it is not clear which number of respondents returned the self-sampling kit (presumably 153) and which part did not (N=80?).

Thank you for this suggestion. We have made significant modifications in the Methods section to provide further details about the participants who participated in the survey and their representativeness of the larger sample of PRESTIS trial participants.

The methods section describes the questions used in the questionnaire, but items like 'telehealth access' cannot be found in the tables or Results section. Moreover, the self-sampling kit is not detailed (vaginal brush/urine sample/…? Which specific device? Which procedure?).

Thank you for noting this discrepancy between the Methods and Results sections. We have eliminate mention of telehealth access in the Methods section as this analysis did not include our results from the telehealth access portion of the telephone survey. We added information in the Methods about the self-sampling device and procedure. We also refer the reader to a published protocol of the parent trial (Montealegre JR, Anderson ML, Hilsenbeck SG, Chiao EY, Cantor SB, Parker SL, Daheri M, Bulsara S, Escobar B, Deshmukh AA, Jibaja-Weiss ML, Zare M, Scheurer ME. Mailed self-sample HPV testing kits to improve cervical cancer screening in a safety net health system: protocol for a hybrid effectiveness-implementation randomized controlled trial. Trials. 2020 Oct 21;21(1):872. doi: 10.1186/s13063-020-04790-5. PMID: 33087164; PMCID: PMC7580009.)

The patient characteristics (demographic data, SES) presented in Table 1 are very limited and specifically 'age' and 'screening history' are lacking (besides 'employment' and 'marital status'), which are important risk factors for studying cervical cancer screening. These data could however easily be obtained during the phone calls, or might even be available from the overall PRESTIS study. Please add if available. These variables would have been very important in bivariate or multivariate analysis to determine the variables associated with HPV self-sampling uptake and the barrier outcomes, to enrich the basic descriptive statistic presented here. Furthermore, the denominator of Table 1 is indicated to be 233, but some variables have missing data, which are not indicated in the table, nor discussed. Specifically, the variable 'household income', being a very important SES factor, was only reported for 136 women. Please report all missing data, and how this could have influenced the distribution shown.

Thank you for this suggestion. We have added age and screening history to Table 1 and agree that these are important variables for understanding the study population.. Table titles now reflect the populations and subgroups that were included in that table. Missing data are indicated as subscripts.

Table 4 should be structured better to make clear to the reader that question 2 only relates to the positive answers in question 1. The denominator shown in the title of this table (N=153) only applies to the first question, the last 3 show data for all 233 interviewed women. Please adjust accordingly.

Table 4 has been restructured to clearly show the N for each group or subgroup for which COVID barriers were assessed.

A more thorough discussion on the overlapping barriers reported in different healthcare systems and the specific deviating barriers – and the reasons why they deviate – in this specific patient group, would have been very informative.

Thank you for this suggestion. We have added to the conclusion and compared our findings with those from other healthcare systems, showing that the importance of certain barriers differs by population (page 15).

Reviewer #1 (Recommendations for the authors):Please find most recommendations in the public review, complete with:(Abstract)– … reported clinic(clinician)-based screening… discomfort seeing male providers.Might be better formulated as: … is experiencing discomfort when seeing male providers.– Do you 'complete' the kit or just 'use' it?

For this analysis “complete” the kit means that the participant self-reported taking their sample and returning it to the health care system.

(Materials and methods)– Can be clarified more.

We added more details here, including the type of swab included in the self-sampling kit. Additional details about the parent trial can be found in the referenced protocol paper (page 4).

– Leave the verbs away in the details on the inclusion criteria for the PRESTIS study.

This has been done (page 4).

(Results)– Given results can be summarized and condensed, and should be enlarged with more in-depth analysis if the data are available.

Thank you for the suggestion. We have summarized and condensed the data reported and added in-depth analysis of additional demographic variables including age, race/ethnicity and education level.

– The percentage of the Mexico-born population should be corrected (39.4% in the text versus 39.7% in Table 1).

Thank you for catching this discrepancy.

(Tables)– Table 1: Give correct denominators for each variable.

Each category has been updated to include any data missing (participant declined to answer) and percentages have been adjusted accordingly. The denominator for each category is now 233 (page 10).

– Table 2: give N for both Spanish- and English-speaking subgroups.

This has been done (page 11).

– Table 4: restructure and give the correct denominator for each question.

This has been done (page 12).

Reviewer #2 (Recommendations for the authors):This manuscript provides important information on barriers to cervical cancer screening participation, and the potential of self-sampling kits to increase participation, especially among ethnic and/or linguistic minority women.The manuscript is clearly and concisely written. The tables lack important information (esp. age of the participating women). P-values in the text are not given as precise values but whether they are below some cutoff; this is not recommended.

Age has been added to Table 1 and precise p-values have replaced the cutoffs (page 10).

The flow of the manuscript can be improved. One of the conclusions ("Furthermore, during the COVID-19 pandemic, these barriers were exacerbated by economic, physical, and mental effects of the pandemic") is not substantiated and could be rephrased.

This has been rephrased to …”these barriers may have been exacerbated…” (page 17). This statement is related to the data presented in Table 4 showing that over half of participants’ economic and mental health was affected by the pandemic and over a third of participants’ physical health was affected, potentially exacerbating existing barriers to cervical cancer screening.

The main conclusion in the Discussion "In conclusion, mailed at-home HPV self-sampling kits present an opportunity to reduce important barriers to cervical cancer screening among women in a safety net healthcare system" stands and is an important message to cervical cancer screening organizations – there is a lot to gain with self-sampling kits.p1, Abstract, and p7 and p8 Results: It is recommended to provide precise P values rather than stating P<0.01 or P<0.05, for all P>0. 0.001.

This change has been made (pages 1, 7-8).

p7, Results: "Significantly more Spanish-speaking participants reported that getting a Pap test is embarrassing compared to non-Hispanic and English-speaking participants, 66.4% vs. 30%, p<0.01" The comparison reported here is correct for "English speaking participants", but was not shown in the analysis for non-Hispanics.

This correction has been made (page 7).

p7, Results: The order of reporting percentages and associations in the 2nd paragraph of Results is different from the order of reporting in Table 2. It is recommended to stick to the same order in the Table and text.

Table 2 has been reordered to reflect the order of Results reported in paragraph 2 (page 11).

p7, Results: be consistent in the number of decimals given for percentages; throughout the Results, one decimal is provided, but in the 5th line from the bottom no decimal is given in "78%".

This correction has been made (page 7).

p7-8, Results, third paragraph: please refer to the appropriate Table here (Table 4).

This correction has been made (pages 7-8).

p8, Results, second paragraph: I'd suggest that the authors first report the number of participants returning the kit before they report analyses restricted to that group, e.g. "153 (xx%) of participants returned the kit."

This correction has been made (page 8).

Results: it is recommended to number the Tables in order of appearance. Now Table 3 is first mentioned after Table 4 is mentioned.

This correction has been made by reordering the Results section.

Tables: it is recommended to provide full Table titles, stating Person, Place, and Time, so that a Table can be more easily understood on its own.

This change has been made.

Table 1: It appears some data are missing (e.g. for Place of birth, Education, Household income). State in the body of the table or in a footnote which number of records had missing values for each variable.

Thank you for this feedback- Table 1 now includes the number of participants who declined to provide their place of birth, education and household income. Percentages have been updated to reflect the prevalence of each with the total N (233) as the denominator (table 10)

Table 1 does not list age; age is crucial for any epidemiological analysis. Please add.

Age has been added to Table 1 (page 10).

Table 2: the order of categories seems somewhat contra-intuitive: with the variable "Uncomfortable with male provider" the first category given is Yes, and the second No. Reverse order?Same for the other variables in this Table 2?

The current order reflects what we have seen in other publications, so this change was not made.

Tables 2 and 3 and 4: Please provide the N above each of the three columns.

This change has been made (pages 11-13).

Table 2: Were P values calculates based on Yes and NO categories only, or also including the Unsure categories?

Also including unsure categories.

It appears "Getting a Pap test is expensive" has a missing value. Please note this in the body of the table or footnote.

This change has been made (page 11).

Table 3: Indicate here in the title who is in the analysis, so as to explain there are only 153 in this Table rather than the 233 participants.

This change has been made (page 12).

Discussion, p14: "Furthermore, during the COVID-19 pandemic, these barriers were exacerbated by economic, physical, and mental effects of the pandemic". I am not sure this has been demonstrated by the analyses the authors report.

This has been rephrased to …”these barriers may have been exacerbated…” This statement is related to the data presented in Table 4 showing that over half of participants’ economic and mental health was affected by the pandemic and over a third of participants’ physical health was affected, potentially exacerbating existing barriers to cervical cancer screening (page 14).

References: Please check the rendering of ref 17.

Reference has been updated to reflect formatting of others (page 20).

Reviewer #3 (Recommendations for the authors):The authors investigated attitudes toward HPV self-sampling as an alternative screening method for under-screened women who live with multiple social disadvantages in an urban area of the USA. Although their study included elements that were specific to the era of the COVID pandemic during which the study was undertaken, it once more confirmed that HPV self-sampling may be an attractive alternative for a proportion of under-screened women. Whether the positive attitudes described in the study would translate into increased screening participation, remains to be seen.

We appreciate these comments. We have added to the discussion to specify that trial results are needed to understand whether positive attitudes translate to increased screening participation.

The authors could provide some more information on how women can seek screening from this health care provider. Does this provider offer cervical cancer screening, and did Arm 1 of the trial (control) actually represent "usual care"? If care is provided regardless of an individual's ability to pay, then why did some women complain that paps are too expensive? Do women need to seek screening themselves, or will they be reminded that they are (over)due? Around 2/3 of the women reported that they are uncomfortable around a male screening provider – how likely is it that a woman will be screened by a male vs. a female provider, and can she choose between the smear-taker's gender?

Thank you for this suggestion. We have added information under Setting to describe usual care cervical cancer screening practices in our health system, as well as augmented usual care provided through the trial. We appreciate the potentially confusion around cost barriers and the safety net health system’s provision of care. We have provided additional information about payment for cervical cancer screening in the *Participants* subsection (page 5). If a woman is determined not to be financially eligible for free care (i.e., she may be determined by the health system to have a plan that requires co-payment or self-payment), out of pocket expenses can still be a burden and barrier to care. While they may have been enrolled in a coverage/financial assistance plan at the time of the survey, the plan may still incur costs.

A question to clarify some of the exclusion criteria: why did women need to have had at least 2 visits with this health care provider in the last 3.5 years? Would this not tend to increase the likelihood that the study would not reach severely underscreened and never-screened women (e.g., those that do not engage with health care but could be motivated to do self-sampling)?

Thank you for pointing out the need to explain the rationale for the 2 visits in the inclusion criteria. Because safety net systems are used to varying extent by population, we instituted eligibility and exclusion criteria to minimize the inclusion of women who use the health system only for emergency, inpatient, and specialty care but otherwise largely receive their primary care elsewhere (or nowhere). We explain in the Methods section that we limited inclusion to women who had visited a primary care clinic at least twice over the past five years and excluded women who had a documented primary care provider outside of the health system in order to capture patients who use the health system for their primary care.

Why was the main analysis looking at English vs. Spanish-speaking participants? It may make sense for policy-making within the specific healthcare provider, but less so for the international audience. Could the authors report some other comparisons e.g., by age or certain other sociodemographic characteristics or the women's screening history? Also, from the exclusion criteria it seems that some women may have been screened only by paps. Did the attitudes towards HPV self-sampling among those women differ from the attitudes among women who were previously screened with co-testing (and indeed from the attitudes among unscreened women, if there were any in the study)? (A simple frequency distribution for such characteristics added to Table 1 would also be helpful)

Thank you for this point. We have added a statement in the limitations paragraph that the main comparison may not be applicable to other health systems or for international audiences (page 16). However, this comparison is highly relevant to our population. Yes, some women were last screened with a Pap, and some with a co-test (the Health System recently switched to co-testing) but this was not included as an analysis since the clinical experience for both tests is nearly identical. Language is often used as a proxy for nativity, citizenship, immigration status and acculturation, so would be very relevant for policy makers during implementation planning for an HPV self-sampling program within the safety net healthcare system. We have added a statement to support this in the introduction. We have also added analyses to Tables 2-4 that include race/ethnicity, age and education level. We did not include income in these additional analyses because of the large number of missing data for this variable.

The authors concluded that self-sampling "may help to address disparities in cervical cancer screening adherence". This is an abstract statement but could be better supported by evidence from their study, for example by reporting how many of the women approached for the trial actually did self-sampling. We know that in theory self-sampling should be able to increase screening participation, but translation into practice has not been extremely successful. That is why statements like these would better not be made unless they can be supported with data.

We agree that this statement cannot be made based solely on the data from this study. We have modified the statement to specify that “If positive attitudes toward self-sampling translate into increased screening participation, self-sampling may help…….”

This analysis of barriers to clinician collection and attitudes around self-sampling comes after more than a decade of intensive research and multiple systematic reviews. The paper should provide a better overview of the existing literature. Several studies have investigated attitudes towards self-sampling and self-sampling uptake in disadvantaged women from high-income countries, which would be consistent with the population in this study, so relevant for comparison.

Thank you, we have added references and context to the Introduction section (page 2).

Finally, the authors stated that their study is the first to describe barriers in obtaining cervical cancer screening during the COVID-19 pandemic (for a specific population). This may be correct, but for most people, whether one likes it or not, life has moved on a while ago. Could the authors find a way to make this paper more interesting (and informative) for a post-pandemic world?

We explain that self-sampling is a potential method to circumvent barriers to cervical cancer screening apart from COVID-related barriers. We’ve added statements to the introduction elaborating on this point and distinguishing COVID-related barriers and barriers that existed pre-pandemic (page 2). However, it’s important to note that the data collected for this analysis was collected at various time points during the pandemic, and the implications of pandemic-related barriers cannot be separated from the data.

Reviewer #4 (Recommendations for the authors):The paper affords an interesting issue and the whole project, including the trial, will timely assess the effectiveness of self-sampling in reducing inequalities in cervical cancer access.It is now time to start implementing self-sampling and try to exploit the potential benefits of this device in increasing HPV test coverage. To do that we need such research because implementation must be context specific. In fact, in this field context-specific, implementation research is needed and results observed in trials conducted in other countries or in different health systems cannot be transferred to a different situation. Furthermore, the acceptability of self-sampling depends on the social and cultural background of the women, but also on how it is offered by the provider and on how the communication between the provider and women occurs.This study will integrate data from a survey and from a trial. In this paper, the authors report the results of the survey. It highlights that part of the barriers reducing access to screening of disadvantaged women can really be removed by self-sampling. The study shows that these barriers are now more important for Hispanic women. These findings are important, but more information from this survey can be exploited.MethodsData collectionI did not understand if all trial participants have been contacted for the interview. Furthermore, I did not understand if the consent was only to participate in the survey or in the trial (it would be a strong limitation to the trial including only those agreeing to participate, even if sometimes Ethics Committee or IRB imposes it).

Thank you for indicating the potential for confusion. We have modified the methods section to clarify who participated in the telephone survey. As stated under Data Collection, “a subset of randomly selected trial participants randomized to home-based self-sample sampling for HPV testing.” In the protocol paper for the original trial, we report that “All eligible women are enrolled in the trial under a waiver of consent in order to reduce participation bias. A waiver of written documentation of informed consent was granted for participants’ use of the kits, due to the minimal risks involved and to enhance generalizability of the findings. A research information letter (described below) is used in lieu of a formal informed consent form.” Verbal consent was obtained for the telephone survey, conducted among a subset of trial participants. All consent procedures were reviewed and approved by Baylor College of Medicine and Harris Health System IRBs.

I think it should be reported a description of the survey power or the sample size determination, whether it has been established on the basis of a formal sample size estimation or with a convenience constraint.MeasuresIt is not clear if the different concepts (embarrassment, discomfort…) are measured in a single question or in separate items.

These are measured in separate items on the survey, which is attached in the original submission. We’ve added a clarifying statement in materials/methods to indicate this (page 5).

ResultsIt is important to report the response rate to the survey. It would be important also to report if the response was differential among those who returned the self-sample (or attended) and those who did not.

The trial is ongoing, so we do not have access to actual kit return data. For the purposes of this analysis, we are relying on self-report of whether a kit was returned. Thus, we cannot assess the return rates of those trial participants who did not respond to, or declined to participate in the telephone survey.

Why did the authors investigate only ethnicity as a possible determinant? Not the age, income, or education? Returning the kit or not? I think such analyses would be relevant to your objectives.

We have added a column in Table 1 to show the breakdown of education and income by language and indicated where significant differences were found (page 10). We have added additional analyses to Tables 2-4 that include race/ethnicity, education and age. We did not include income due to the large number of missing data for this variable.

DiscussionIt is balanced and well-written.I suggest discussing the limitations in light of other results, not in a separate paragraph.

We’ve added comparisons to other results, and have kept general limitations in a separate paragraph.